# Hesperetin, a Promising Dietary Supplement for Preventing the Development of Calcific Aortic Valve Disease

**DOI:** 10.3390/antiox11112093

**Published:** 2022-10-24

**Authors:** Hengli Zhao, Gaopeng Xian, Jingxin Zeng, Guoheng Zhong, Dongqi An, You Peng, Dongtu Hu, Yingwen Lin, Juncong Li, Shuwen Su, Yunshan Ning, Dingli Xu, Qingchun Zeng

**Affiliations:** 1State Key Laboratory of Organ Failure Research, Department of Cardiology, Nanfang Hospital, Southern Medical University, Guangzhou 510515, China; 2Guangdong Provincial Key Laboratory of Shock and Microcirculation, Southern Medical University, Guangzhou 510515, China; 3Bioland Laboratory (Guangzhou Regenerative Medicine and Health Guangdong Laboratory), Guangzhou 510005, China; 4School of Laboratory Medicine and Biotechnology, Southern Medical University, Guangzhou 510515, China; 5Department of Cardiovascular Surgery, Nanfang Hospital, Southern Medical University, Guangzhou 510515, China; 6Division of Obstetrics and Gynecology, Nanfang Hospital, Southern Medical University, Guangzhou 510515, China

**Keywords:** hesperetin, calcified aortic valve disease, Sirt7, Nrf2

## Abstract

Background: No effective therapeutic agents for calcific aortic valve disease (CAVD) are available currently. Dietary supplementation has been proposed as a novel treatment modality for various diseases. As a flavanone, hesperetin is widely abundant in citrus fruits and has been proven to exert protective effects in multiple diseases. However, the role of hesperetin in CAVD remains unclear. Methods: Human aortic valve interstitial cells (VICs) were isolated from aortic valve leaflets. A mouse model of aortic valve stenosis was constructed by direct wire injury (DWI). Immunoblotting, immunofluorescence staining, and flow cytometry were used to investigate the roles of sirtuin 7 (Sirt7) and nuclear factor erythroid 2-related factor 2 (Nrf2) in hesperetin-mediated protective effects in VICs. Results: Hesperetin supplementation protected the mice from wire-injury-induced aortic valve stenosis; in vitro, hesperetin inhibited the lipopolysaccharide (LPS)-induced activation of NF-κB inflammatory cytokine secretion and osteogenic factors expression, reduced ROS production and apoptosis, and abrogated LPS-mediated injury to the mitochondrial membrane potential and the decline in the antioxidant levels in VICs. These benefits of hesperetin may have been obtained by activating Nrf2–ARE signaling, which corrected the dysfunctional mitochondria. Furthermore, we found that hesperetin could directly bind to Sirt7 and that the silencing of Sirt7 decreased the effects of hesperetin in VICs and potently abolished the ability of hesperetin to increase Nrf2 transcriptional activation. Conclusions: Our work demonstrates that hesperetin plays protective roles in the aortic valve through the Sirt7–Nrf2–ARE axis; thus, hesperetin might be a potential dietary supplement that could prevent the development of CAVD.

## 1. Introduction

Calcific aortic valve disease (CAVD) affected approximately 12.6 million patients and caused over 10,000 fatalities in 2017 [1]. It is characterized by a progressive calcification and fibrosis of the aortic valve leaflets, and progressive calcification and fibrosis will further lead to the obstruction of the left ventricular outflow and ultimately induce aortic stenosis [2]. Characteristics of CAVD have been idetified previously, which included the early involvement of inflammation and apoptotic vesicles [3], and the transition of the cellular phenotype as aortic valve interstitial cells (VICs) differentiate into myofibroblasts [4,5] in response to reactive oxygen species (ROS)-mediated oxidative stress (OS) [6]. However, since no effective pharmacological therapies are available, CAVD causes a high clinical and economic burden [7].

Hesperetin, a flavanone [8], is abundant in oranges, lemons, and other citrus fruits. Numerous investigations demonstrate that hesperetin performs various biological activities in the cardiovascular system and neurodegenerative diseases, including its anti-inflammatory, antioxidant, and antiapoptotic effects [9,10]. In addition, hesperetin has been applied as a novel formulation in health supplements and drinks to combat aging and improve cardiovascular health. However, the role of hesperetin in CAVD has not been clarified.

Nuclear factor erythroid 2-related factor 2 (Nrf2), a master transcription factor, regulates the cellular redox homeostasis by activating antioxidant response element (ARE)-responsive genes, including NAD(P)H-quinone oxidoreductase 1 (NQO1) and heme oxygenase-1 (Hmox-1) [11]. Numerous studies have reported that upregulating the Nrf2 system can attenuate high phosphorus-induced calcification in vascular smooth muscle cells [12], and the activated Nrf2/HO-1 signaling axis can suppress the calcification of VICs [13]. For these reasons, targeting the Nrf2/HO-1 pathway may be a promising therapeutic strategy for suppressing inflammation and OS and ultimately inhibiting valve calcification. Notably, current study now demonstrates that hesperetin can ameliorate inflammation and OS [14], rescue lipopolysaccharide (LPS)-induced apoptosis [15] through activating the Nrf2 pathway.

The sirtuin family was originally identified as a series of longevity-related genes. To date, seven sirtuin genes have been identified, and sirtuin 7 (Sirt7) is the most recent identified mammalian sirtuin [16]. Recent studies have shown that Sirt7, a class III histone deacetylase, hats deacetylase, desuccinylase, and deglutarylase activities [17] and plays various roles in regulating gene transcription and chromatin structure, activating DNA repair, and in metabolic adaptation [18]. Additionally, Sirt7 deficiency can induce mitochondrial dysfunction to result in increased cardiomyocyte apoptosis and inflammation [16,19]. Although inflammation, OS, and apoptosis are closely related to valve stenosis and calcification, sirt7 function in this process is still poorly understood. Importantly, a large body of literature suggests that the sirtuin family is an upstream regulator of Nrf2 pathway. However, the direct link between Sirt7 and Nrf2 has not been explored in CAVD or other cardiovascular diseases (CVDs).

Currently, lifestyle interventions, including a normal diet and exercise training, have limited effects on slowing the progress of CAVD once it occurs [20], and effective medical treatment for CAVD is still lacking. Therefore, dietary food supplements may be a promising emerging therapeutic strategy for CAVD.

Here, we tested the hypothesis that hesperetin supplementation exerts a protective effect in CAVD. To this end, we interrogated the effect of hesperetin on the extent of valve thickening and the aortic valve peak velocity function in mice with direct wire injury (DWI), explored the effect of hesperetin on LPS-induced inflammatory responses, OS, and apoptosis, and found that LPS-induced inflammation and apoptosis in VICs are dependent on ROS. We further investigated the mechanism of hesperetin-mediated beneficial effects and ultimately demonstrate that hesperetin exerts anti-inflammatory, antioxidant, and antiapoptotic effects via the Sirt7–Nrf2–ARE axis.

## 2. Methods

### 2.1. Primary Cell Isolation and Culture

Specific details regarding human VICs isolation has been described in our published paper [21].

The effect of hesperetin on the inflammatory responses under LPS stimulation were determined by pretreating human VICs with or without hesperetin (3 µM) for 24 h and then treated with LPS (200 ng/mL, Sigma, L4391) for another 24 h.

To determine the influence of hesperetin on the LPS-induced osteogenic differentiation, we pretreated human VICs with or without hesperetin (3 µM) for 24 h and then treated with LPS (200 ng/mL) for 3–28 days.

The effect of hesperetin on the apoptosis under LPS stimulation were determined by pretreating human VICs with or without hesperetin (3 µM) for 24 h and then treated with LPS (1000 ng/mL) for 4–24 h.

To determine the influence of hesperetin on the LPS-induced OS, human VICs were pretreated with or without hesperetin (3 µM) for 24 h and then treated with LPS (200 ng/mL) for another 24 h.

To explore whether Sirt7 is involved in the regulation of hesperetin on human VICs, VICs were transfected with Sirt7 small interfering RNA (siRNA) (20 nM, OBiO) or Scrambled siRNA for 24 h and then stimulated with LPS with or without hesperetin (3 µM).

### 2.2. Animal Model

All 8-week-old male C57BL/6 mice used in this experiment were purchased from the Laboratory Animal Center of Southern Medical University. Specific details regarding Aortic aortic stenosis mouse model have been previously described. To be brief, aortic stenosis is induced by directly inserting a wire into the left ventricle and then repeatedly piercing the aortic leaflets 30 times with the body of the wire [21,22]. Sham-treated mice received all surgical procedures but without wire insertion into the left ventricle. 

All the mice were randomly divided into four groups. One group is to evaluate the preventive effects of hesperetin, started one week before DWI surgery, with hesperetin (MedChemExpress, Monmouth Junction, NJ, USA) suspended in water containing 0.1% carboxymethyl cellulose (MedChemExpress), administered daily via oral gavage. One group was administered daily with carboxymethyl cellulose (CMC-Na), started one week before DWI surgery, and the other two groups were sham and sham + CMC-Na (Figure 1A). Aortic valve velocity was measured by echocardiography at 8 weeks after surgery. And we then sacrificed the mice and collected the hearts.

### 2.3. Cell Viability

Specific details regarding cell viability has been described in our published paper [21]. To be brief, we added human VICs to 96-well plates, stimulated VICs with hesperetin (0-50 µM) for 24h and then examined by using the Cell Counter Kit 8 (CCK-8) assay (DOJINDO, Kumamoto, Japan). 

### 2.4. Alizarin Red Staining

After 4 weeks of LPS stimulation, alizarin red staining was carried out to assess the calcium deposition based on the manufacturer’s instructions. To be brief, cells were fixed in 4% paraformaldehyde for 10 minutes after washed with PBS 3 times and then stained with alizarin red solution for 30 min. Distilled water were used to clean the excess stain solution.

### 2.5. ROS and MCMP Detection

Specific details regarding ROS and MCMP Detection were described in our published paper [21]. Briefly, ROS was examined by using 2,7-dichlorodihydro-fluorescein diacetate (DCFH)-DA (Beyotime, S0033S Haimen, China) and the MCMP was detected by using the JC-1 Kit and tetramethylrhodamine (TMRM) Kit (Beyotime, C2001S). Images were captured using a Leica TCS SP8 (Wetzlar, Germany) and analyzed using ImageJ software (NIH, Bethesda, MA, USA).

### 2.6. Cell Transfection

Human VICs were transfected with 50 nM siRNA targeting human Sirt7 (Tsingke Biotechnology Co., Beijing, China) once the cell density reached 70%. Lipofectamine 3000 (Invitrogen, Waltham, MA, USA) were used as the transfection reagent in our experiment.

### 2.7. Immunoblotting

Specific details regarding protein extraction were described in our published paper [21]. The primary detection antibodies used in our experiment were anti-beta-actin (1:1000, Proteintech, Rosemont, IL, USA), anti-Sirt7 (1:1000, Santa, Dallas, TX, USA), anti-alkaline phosphatase (ALP) (1:1000, ABclonal, Wuhan, China), anti-MCP-1 (1:1000, ABclonal, Wuhan, China), anti-Nrf2 (1:1000, Proteintech, Rosemont, IL, USA), anti-NQO1 (1:1000, Proteintech, USA), anti-Hmox1 (1:1000, Proteintech, Rosemont, IL, USA), anti-ICAM-1 (1:1000, Abcam, Waltham, MA, USA), anti-BAX (1:1000, CST, Boston, MA, USA), anti-runt-related transcription factor 2 (Runx2) (1:1000, Proteintech, Rosemont, IL USA) and anti-cleaved caspase3 (1:1000, CST, Boston, MA, USA). The secondary antibodies used were goat anti-rabbit and anti-mouse IgG-HRP (1:5000, Fudebio, Hangzhou, China). The quantitation of Immunoblotting images was analyzed using ImageJ software (NIH, Bethesda, MA, USA).

### 2.8. Immunofluorescence Staining

Immunofluorescence staining was used to examine the ICAM1, HO-1, Sirt7, cleaved-caspase3, and Runx2 levels in the aortic valves of mice, and we also examined the levels of Sirt7 in calcified aortic valves and noncalcified aortic valves in a human model. Specific details regarding immunofluorescence staining approaches were described in our published paper [21]. The quantitation of immunofluorescence staining images was analyzed using ImageJ software (NIH, Bethesda, MA, USA).

### 2.9. Flow Cytometry

Cell apoptosis was examined using FITC Annexin V Apoptosis Detection Kit I (Biosciences-556547, Franklin Lakes, NJ, USA) according to the manufacturer’s protocols. Human VICs were seeded into P6 culture dishes and treated with LPS (1 µM) and hesperetin (3 µM) for 4 h. The results were analyzed using Cyan ADP 9C flow cytometer (Beckman Coulter, Imagoseine platform, Institute Jacques Monod, Paris, France). We performed three analyses on three independent cell samples in each condition and each analysis was based on at least 50 000 events. In addition, the whole experiment repeated independently at least two times.

### 2.10. Pathway and Functional Enrichment Analyses

Pathway and functional enrichment analyses were performed using the Kyoto Encyclopedia of Genes and Genomes (KEGG) and Gene Ontology (GO) analyses by using R version 4.0.4 (R Foundation for Statistical Computing, Vienna, Austria).

### 2.11. Statistical Analysis

All data in this study are presented as means ± SEMs. The significant differences between two or more than two groups were determined by a two-tailed unpaired Student’s *t*-test and one-way ANOVA, followed by the Tukey post hoc test, respectively. All statistical analyses were performed using GraphPad Prism version 8.0.

## 3. Results:

### 3.1. Hesperetin Supplementation Protects the Mice from Wire Injury-Induced Aortic Valve Stenosis 

Previous studies have demonstrated that hesperetin plays a role in protecting against myocardial ischemia and many other CVDs [23]. To evaluate the role of hesperetin in attenuating aortic stenosis, we adopted a new aortic valve stenosis model by by directly inserting a wire into the left ventricle and repeatedly piercing the aortic leaflets [22]. One week before the procedure, hesperetin was administered at a dose of 50 mg/kg [24] daily (Figure 1A). Eight weeks after injury, the echocardiographic showed that the aortic valve peak velocity increased significantly in DWI group (Figure 1B), compared with those of the sham and sham + CMC-Na groups, and hesperetin supplementation reversed these changes (Figure 1B–D). As expected, hematoxylin and eosin (HE) staining, which was applied to observe the morphology of the valve leaflet, showed that the thickness of leaflet was greater in the wire-injury group. In contrast, aortic valve thickness was reduced in DWI+Hst group (Figure 1C). Next, we evaluated the degree of calcification, and immunofluorescence staining showed that mice subjected to the DWI surgery exhibited a increased levels of the Runx2 (a classic osteoblastic differentiation protein marker [25]) in the aortic valve, but this increase could be markedly attenuated by hesperetin supplementation (Figure 1D). The above evidence indicates that hesperetin supplementation ameliorates aortic stenosis and calcification in the wire-injury mice.

### 3.2. Hesperetin Supplementation Alleviates Inflammation Caused by DWI In Vivo or LPS-Induced Inflammation In Vitro

To reveal the specific potential molecular targets and pathways related to hesperetin, we utilized the Traditional Chinese Medicine Systems Pharmacology (TCMSP) and National Library of Medicine (NLM) databases. As shown in Figure 2A, the KEGG analysis [26] revealed that NF-κB signaling pathways are strongly related to the cytoprotective mechanism of hesperetin. Therefore, we evaluated the effect of hesperetin on inflammation. We first examined the expression of ICAM1, a cell-surface glycoprotein that can regulate leukocyte recruitment from the circulation to sites of inflammation [27] in vivo. 

Immunofluorescence revealed that mice subjected to the DWI surgery exhibited a increased levels of the ICAM1 in the aortic valve compared with those in the sham and sham + CMC-Na groups, but this increase was markedly attenuated by hesperetin supplementation (Figure 2B). The chemical structure of hesperetin is shown in Figure 2C.

VICs, the primary cell type found within heart valves, stabilize the microenvironment under normal physiological conditions and regulate the progression of valve disease [28]. Mounting evidence suggests that inflammatory dysregulation causes increased valvular calcification under LPS-stimulated conditions [29]. Thus, we assessed the potential function of hesperetin in LPS-treated VICs.

We first assessed the effect of hesperetin on VIC viability. The results of MTT assays showed that there was no clear inhibitory effect on the growth of VICs exist under hesperetin (0–50 µM) treatment (Figure 2D). Next, we examined the hesperetin treatment alone in VICs and found that hesperetin might exert protective effects on VICs, which originated from CAVD patients but not on healthy donors (Appendix A). To further clarify the role of hesperetin, we then investigated the inflammation-related changes in human VICs and found that the levels of inflammatory genes (ICAM-1, MCP-1, and p-p65), an inflammatory cytokine (IL-6), and a chemokine (RANTES) were higher after LPS stimulation (200 ng/mL, 24 h) than those of the control groups (Figure 2F,G). In contrast, the VICs pretreated with hesperetin (3 µM) for 24 h and stimulated with LPS for another 24 h presented significantly reduced levels of these genes. Furthermore, immunofluorescence staining confirmed the augmented translocation of active p65/NF-κB into the nucleus under LPS treatment. This translocation was abolished by hesperetin supplementation (Figure 2E).

We further explored the effects of hesperetin on the osteogenic response. Immunoblotting showed that LPS-treated VICs had higher levels of Runx2 and ALP, while pretreatment with hesperetin inhibited osteogenic differentiation (Figure 2H). In addition, Alizarin red staining showed that hesperetin supplementation eventually reduced LPS-induced calcium deposition (Figure 2I).

Taken together, these results suggested that hesperetin supplementation can alleviate DWI-induced inflammatory response in vivo and inhibit LPS-induced osteogenic differentiation and inflammation in vitro.

### 3.3. Hesperetin Supplementation Alleviates LPS-Induced Apoptosis in VICs and DWI-Induced Apoptosis In Vivo

As shown in Figure 3A, GO analysis [30] revealed that the potential molecular targets of hesperetin are mainly associated with the regulation of apoptosis and the response to LPS (Figure 3A). Therefore, we investigated the effect of hesperetin on the apoptosis of VICs. As shown in Figure 3B, immunofluorescence staining showed that the levels of cleaved-caspase3 were increased in the DWI + CMC-Na group, but this increase was reversed in the hesperetin supplementation group.

In vitro experiments showed that LPS-treated VICs had higher protein levels of cleaved-caspase3, caspase3, and BAX than control VICs, and these increased protein levels were profoundly decreased in the LPS + Hst group (Figure 3C and Appendix A). Furthermore, flow cytometry and immunofluorescence staining showed that hesperetin attenuated LPS-induced apoptosis in VICs (Figure 3D,F).

### 3.4. Hesperetin Supplementation Alleviates LPS-Induced OS in VICs and DWI-Induced Apoptosis In Vivo

Previous studies have shown that hesperetin ameliorates OS [14]. Thus, we investigated the effects of hesperetin in a wire injury model. Immunofluorescence showed that the levels of HO-1 were decreased in the DWI + CMC-Na group, but hesperetin supplementation normalized the HO-1 levels (Figure 4A). We also analyzed the effect of hesperetin supplementation on LPS-induced OS. As expected, we found that hesperetin supplementation markedly attenuated LPS-induced ROS production in human VICs (Figure 4B). A previous study demonstrated that the MCMP is a critical indicator of mitochondrial function, and a decreased MCMP is related to an increase in mitochondrial ROS production [31]. Thus, the MCMP was examined by JC-1 and TMRM staining, which showed that the MCMP was higher in LPS + Hst group than LPS alone group in VICs (Figure 4C,D), and this finding suggests that hesperetin attenuates mitochondrial dysfunction. Immunoblotting was then performed to examine the levels of antioxidant proteins (HO-1 and NQO1) in VICs pretreated with 3 µM hesperetin for 24 h. LPS-exposed cells showed decreased HO-1 and NQO1 levels, but these decreases were reversed by hesperetin treatment (Figure 4E). Thus, the above results indicated that hesperetin supplementation improved the ability of VICs to mitigate the LPS-induced OS by upregulating antioxidant proteins.

### 3.5. LPS-Induced Inflammation and Apoptosis in VICs Are ROS-Dependent

Growing evidence indicates that ROS can trigger apoptosis and inflammation [32]. Thus, we hypothesized that the induction of inflammation and apoptosis in VICs induced by LPS was dependent on ROS. VICs were treated with LPS and 100 µM N-acetyl-L-cysteine (NAC), a classic antioxidant, for 24 h. As shown in Figure 4F, NAC treatment significantly inhibited the upregulation of ICAM-1, MCP-1, BAX, and cleaved-caspase3 expression induced by LPS. These results suggested that LPS induces inflammation and apoptosis in VICs are ROS-dependent.

### 3.6. Hesperetin Supplementation Upregulates Nrf2–ARE Signaling

Activation of Nrf2 pathway has a protective effect inflammation and OS [33]. To further determine the role of Nrf2 in the effects of hesperetin, we examined the protein levels of Nrf2 in VICs. As shown in Figure 5A,B, compared with the LPS-treated group, hesperetin supplementation showed an increase in the abundance of Nrf2.

To further determine whether hesperetin can activate Nrf2, we then applied ML385, a specific Nrf2 inhibitor, to inhibit the activity of Nrf2. We pretreated VICs with ML385 for 48 h and then stimulated the cells with hesperetin. ML385 treatment for 24 h essentially eliminated the activity of Nrf2 (p-Nrf2); however, another group stimulated with ML385 + Hst showed no significant difference in the protein levels of p-Nrf2, compared with those of the control group (Figure 5D). 

Immunofluorescence staining was then performed to examine the distribution of Nrf2 in VICs supplemented with hesperetin. Hesperetin supplementation augmented the accumulation of Nrf2 within the nucleus, compared with that in the group treated with ML385 alone (Figure 5C).

Furthermore, we investigated whether hesperetin attenuates ML385-induced inflammation, OS, or apoptosis. We found that the pretreatment of VICs with hesperetin restored the protein levels of HO-1, and NQO1, which were suppressed by prior ML385 stimulation (Figure 5E). Similarly, pretreatment with hesperetin significantly decreased the protein levels of MCP-1, BAX, and ALP, which were increased by ML385 (Figure 5F and Appendix A). Therefore, these results suggested that hesperetin exerts its protective effect by upregulating Nrf2–ARE signaling.

### 3.7. Sirt7 Is Involved in the Protective Effects of Hesperetin in VICs

Numerous reports in the literature have shown that the sirtuin family is an upstream regulator of Nrf2 [34,35,36]. Therefore, we examined all sirtuin factors (Appendix A) and found that only Sirt7, reduced by prior LPS stimulation, was significantly increased in abundance under hesperetin supplementation. Previous studies have demonstrated that Sirt7 can regulate inflammation, OS, and apoptosis, and its beneficial effects have been reported in the vascular endothelium and many other cardiovascular models [37,38,39]. However, the role of Sirt7 in CAVD is incompletely understood, and the relationship between Sirt7 and Nrf2 remains undefined.

To further investigate whether Sirt7 is involved in the mechanism of CAVD, we first examined the expression levels of Sirt7 in patients with CAVD and non-CAVD. Immunofluorescence staining showed that the levels of Sirt7 were lower in CAVD patients than the non-CAVD patients (Figure 6A). Western blotting revealed the same conclusion: The protein levels of Sirt7 were lower in the calcified aortic valves of CAVD patients than in the aortic valves of non-CAVD subjects (Figure 6B). In addition, we also examined the Sirt7 levels in the DWI group and the sham group and reached the same results (Appendix A). Next, through computational docking, we found that hesperetin occupies the active sites of Sirt7, with a binding energy of −8.6 kcal/mol (Figure 6C). Furthermore, Western blotting showed that the LPS-exposed cells exhibited decreased Sirt7 levels, but this decrease was reversed by hesperetin supplementation (Figure 6D).

### 3.8. Sirt7 Modulates Nrf2 Activation in VICs

To further explore the relationship between Sirt7 and the Nrf2–ARE pathway, we used specific siRNAs to knock down Sirt7. VICs were pretreated with Sirt7 siRNA or control scrambled siRNA for 24 h, and then treated with LPS and hesperetin for 24 h. We first examined the effects of hesperetin alone, siRNA scramble treatment alone, and siRNA Sirt7 treatment alone on the VICs (Appendix A). As shown in Figure 7A, the silencing of Sirt7 decreased the protein expression of Nrf2, even in the presence of hesperetin. In contrast, the addition of hesperetin to the VICs transfected with scrambled siRNA increased the levels of Nrf2. Moreover, Sirt7 knockdown enhanced the changes in the BAX, cleaved-caspase3, caspase-3, MCP-1, and ICAM-1 levels in response to LPS in VICs (Figure 7B,C and Appendix A) and suppressed the changes in the NQO1 and HO-1 levels in response to LPS (Figure 8A). In addition, immunofluorescence staining showed that hesperetin supplementation prevented the translocation of active p65/NF-κB into the nucleus under LPS-stimulated conditions, but this effect was abrogated by Sirt7 siRNA (Figure 7D). An analysis of apoptosis through flow cytometry revealed no significant change in the response of VICs to LPS stimulation after Sirt7 knockdown with or without hesperetin supplementation (Figure 7E).

We subsequently observed the impact of Sirt7 on the changes in the MCMP and cellular ROS levels induced by LPS and hesperetin supplementation. Immunofluorescence staining showed that hesperetin treatment significantly decreased the ROS levels and increased the MCMP levels under LPS-stimulated conditions. Conversely, Sirt7 siRNA further upregulated the LPS-induced production of ROS, decreased the MCMP (Figure 8B,C), and eventually aggravated LPS-induced osteogenic response (Appendix A). In summary, these results verified that Sirt7 is involved in the effects of hesperetin through modulating Nrf2 activation in VICs.

## 4. Discussion

Chronic inflammation, a crucial factor, contributes to the initiation and progression of CAVD [40]. New insights have suggested that inflammation precedes valve calcification and increases the levels of cell adhesion molecules such as ICAM1 and VCAM1 [41], which are responsible for recruiting inflammatory cells such as monocytes and macrophages into the valvular tissue, and this increased expression ultimately results in the production of proinflammatory mediators that regulate osteogenic differentiation and calcification in aortic VICs [42]. Similarly, our latest study showed that the inflammatory response of VICs was enhanced by monocytes through by monocytes through binding the secreted factor integrin to ICAM1 [43]. Therefore, the inhibition of inflammation is an important measure to suppress the osteogenic response of aortic VICs. In line with previous findings, in this study, the VICs stimulated with LPS showed significantly higher levels of the inflammatory markers MCP-1, ICAM-1, and p-p65. Moreover, the in vivo experiments showed that the expression of the inflammatory factor ICAM-1 was higher in thickened valves. Recent studies have also proposed that the accumulation of damaging ROS causes oxidation and inflammation in both the initiation phase and the propagation phase of CAVD by augmenting the expression of vascular and intercellular adhesion molecules and inducing the transition of VICs into myofibroblasts and osteoblasts [6]. This differentiation of VICs is induced by the upregulation of osteogenesis genes, including ALP and Runx2 [6]. The differentiated VICs then secrete microvesicles containing ectonucleotidases such as ALP and thus promote calcium phosphate nucleation within valve leaflets. This process was confirmed in our studies, which showed that the stimulation of VICs with LPS increased calcium deposition (Figure 2I) and the levels of ROS (Figure 4B) and decreased the MCMP (Figure 4C,D). The expression levels of the antioxidant proteins HO-1 and NQO1 was also suppressed in the wire injury-induced mouse model (Figure 4A) or under LPS stimulation (Figure 4E). In addition, the apoptotic bodies released from VICs serve as sites of calcium and phosphorous crystal deposition [44,45]. Thus, it is possible that the pathological processes of the inflammatory response and apoptosis in CAVD are dependent on ROS reactions. Indeed, we found that cleaved-caspase3 was strongly expressed in thickened valves (Figure 3B), and LPS increased the expression of cleaved-caspase3 and BAX (Figure 3C). Moreover, we treated VICs with NAC, a classic antioxidant, and the ability of LPS to promote apoptosis and inflammation was abrogated. Together, these observations suggest that LPS-induced inflammation and apoptosis in VICs occur in a ROS-dependent manner (Figure 4F) and that the inhibition of OS is a promising strategy for CAVD.

Numerous studies have demonstrated an inverse association between flavanone intake and the risk of CVD events [46]. Hesperetin, a member of the flavanone family, is abundant in oranges, lemons, and other citrus fruits. Notably, hesperidin, the food-binding form of hesperetin, is the most common flavonoid monomer in the European diet and one of the main components of the traditional Chinese medicine Chenpi [47]. Previous studies have reported that hesperetin exerts diverse protective effects involving the normalization of glucose metabolism, the suppression of proinflammatory cytokine production, and the inhibition of OS [46,47]. Our results showed that pretreatment with hesperetin could reverse LPS-induced proinflammatory cytokine production and osteogenesis in vitro (Figure 2E–I) and prevent DWI-induced valve stenosis and calcification in vivo (Figure 1A,D). Due to the antioxidative and antiapoptotic properties of hesperetin, in vivo experiments showed that HO-1 expression was markedly increased. Additionally, the cleaved-caspase3 levels were increased in the CAVD mouse model, but exogenous supplementation with hesperetin normalized the expression of both HO-1 and cleaved-caspase3. Furthermore, the in vivo results were supported by the results of in vitro experiments, which showed that hesperetin pretreatment suppressed the LPS-induced increase in apoptotic markers (cleaved-caspase3 and BAX) and reversed the LPS-induced decrease in HO-1 and NQO1 levels. Mechanistically, our data obtained with human VICs demonstrated that hesperetin treatment increased the levels and activation of Nrf2, which is consistent with the results reported by Li et al. [14]. In particular, we noted a decrease in NQO1 and HO-1 (Figure 5E) and increases in MCP-1 and BAX (Figure 5F) in VICs when Nrf2 activation was inhibited by ML385, whereas treatment with hesperetin blocked the ML385-mediated inhibition of Nrf2 activation. These results corroborate the indispensable role of Nrf2 in the beneficial effects of hesperetin.

Sirt7 deficiency induces multisystemic mitochondrial dysfunction, and its targeted activation improves mitochondrial homeostasis in CVDs [16,19]. Hence, Sirt7 is considered a potential antioxidant gene and a promising target in CAVD therapy. In human valve tissue, Sirt7 expression was markedly reduced in CAVD patients, compared with non-CAVD patients (Figure 6A,B), and the same result was obtained in the DWI-induced valve thickening (Appendix A). In addition, hesperetin supplementation restored the decreased levels of Sirt7 under LPS stimulation. Furthermore, our docking analysis provided direct evidence showing that hesperetin may bind Sirt7. Therefore, we believe that hesperetin exerts multiple protective effects in CAVD by increasing the Sirt7 expression. Indeed, our data, shown in Figure 7 and Figure 8, support this hypothesis; in our system, the knockdown of Sirt7 with siRNA blocked the hesperetin-mediated inhibition of ROS production, inflammatory cytokine release, and apoptosis under LPS stimulation. In addition, the increases in the antioxidant protein expression and activity induced by hesperetin were clearly suppressed in the presence of Sirt7 siRNA. Regarding the relationship between Sirt7 and Nrf2, numerous studies have shown that the sirtuin family is an upstream regulator of Nrf2 [48,49]. Therefore, we examined all sirtuin factors and found that only Sirt7 showed a significant increase in abundance under hesperetin treatment (Appendix A). We speculate that hesperetin activates Nrf2 by upregulating Sirt7. Importantly, we found that the ability of hesperetin to increase the Nrf2 protein levels was abolished after the inhibition of Sirt7. Collectively, these results suggest that hesperetin exerts its protective effects by regulating the levels of Sirt7 and increasing the Sirt7-mediated activation of the Nrf2–ARE axis.

The present study has several limitations. First, this study examined only aortic VICs. However, endothelial cells may also be involved in aortic valve calcification [50], further study of hesperetin is still needed to determine whether it plays a functional role in valve endothelial cells. Second, we only demonstrated that hesperetin activates Nrf2 by increasing the expression of Sirt7, but the exact mechanism through which Sirt7 regulates Nrf2 has not yet been revealed.

## 5. Conclusions

Dietary supplements have emerged as a new therapeutic strategy. Our in vitro and in vivo experiments demonstrated that hesperetin plays multiple protective roles in the aortic valve through the Sirt7–Nrf2–ARE axis. Therefore, hesperetin, a recent reported dietary supplement, could be a potential therapeutic strategy for preventing the development of CAVD.

## Figures and Tables

**Figure 1 antioxidants-11-02093-f001:**
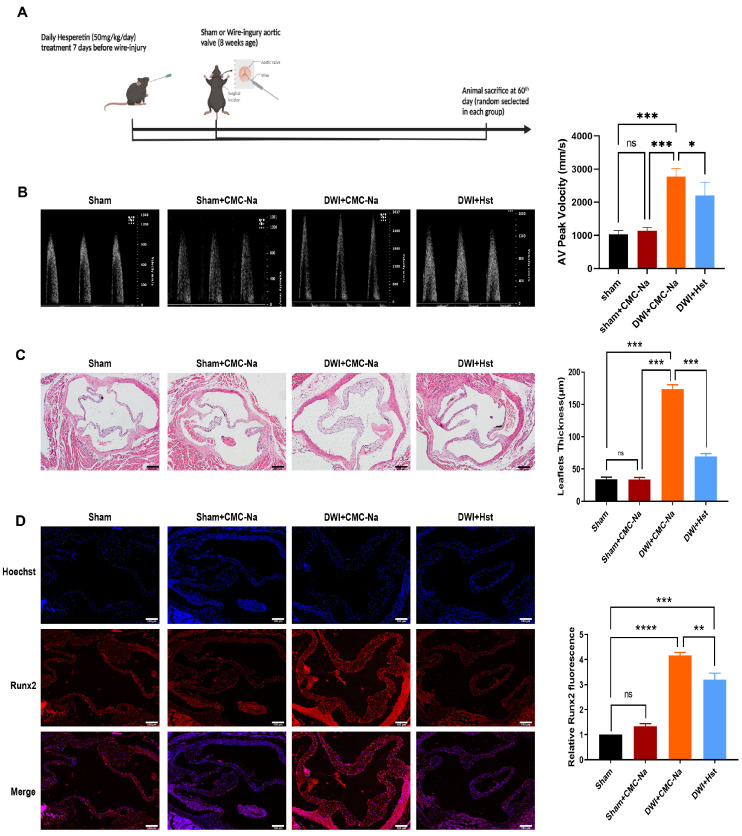
Hesperetin supplementation protects the mice from wire-injury-induced aortic valve stenosis. (**A**) mice treated with direct wire injury or sham operation were administered vehicle (CMC-Na) or hesperetin daily with oral gavage for 7 days before wire injury surgery and sacrificed on the 60th day; (**B**) representative echocardiogram of aortic valve peak velocity; (**C**) hematoxylin and eosin staining (HE) of aortic valve leaflets; scale bar = 200 μm; *n* = 5; (**D**) the Runx2 protein levels (red) in mouse aortic valves were detected by immunofluorescence staining, nuclear staining was performed with Hoechst (blue); scale bar = 100 μm; *n* = 5; * indicates *p* < 0.05, ** indicates *p* < 0.01, *** indicates *p* < 0.001, **** indicates *p* < 0.0001, and ns indicates *p* > 0.05.

**Figure 2 antioxidants-11-02093-f002:**
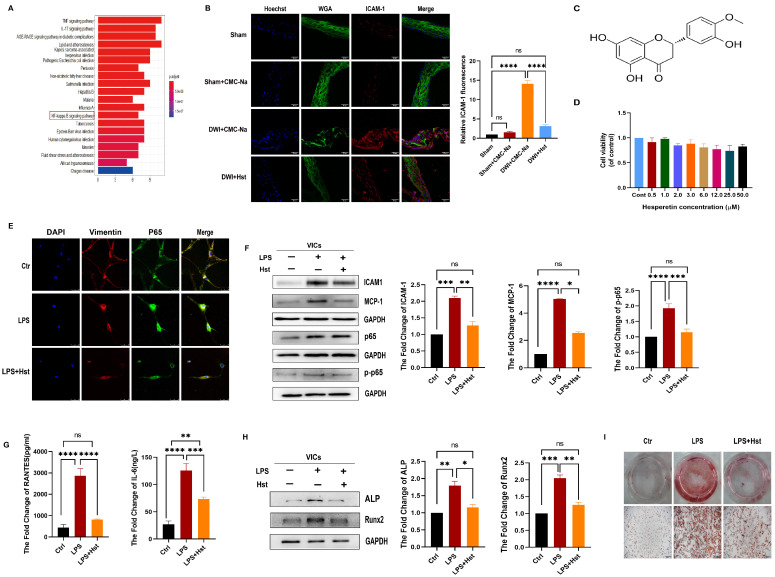
Hesperetin supplementation alleviates inflammation caused by DWI in vivo or lipopolysaccharide (LPS)-induced inflammation in vitro. (**A**) KEGG analysis showing that the effect of hesperetin is related to the NF-κB signaling pathway; (**B**) the ICAM-1 protein levels (red) in mouse aortic valves were detected by immunofluorescence staining, nuclear staining was performed with Hoechst (blue), and WGA (green) was used to show the morphology of the valve leaflet; scale bar = 50 μm; *n* = 5; (**C**) molecular structure of hesperetin; (**D**) cell viability after treatment with different concentrations of hesperetin for 24 h (0–50 μM), *n* = 5; (**E**) immunofluorescence staining was performed to confirm that hesperetin inhibits the LPS-induced cell nuclear translocation of P65 (green), nuclear staining was performed with DAPI (blue), and Vimentin (green) was used to show the morphology of the VICs; scale bar = 50 μm; *n* = 5; (**F**) VICs were pretreated with or without 3 μM hesperetin for 24 h and then stimulated with or without LPS for 24 h, and the protein expression levels of ICAM-1, MCP-1, and p-p65 were determined by Western blotting; *n* = 5; (**G**) the levels of IL-6 and RANTES were detected by using ELISA, *n* = 5; (**H**) under LPS stimulation for 3d, the levels of ALP and Runx2 were detected by immunoblotting; *n* = 3; (**I**) Alizarin Red S staining of human aortic VICs under 3 conditions (control, LPS, and LPS + Hst) for 4 weeks; scale bar = 500 μm; *n* = 5. * indicates *p* < 0.05, ** indicates *p* < 0.01, *** indicates *p* < 0.001, **** indicates *p* < 0.0001, and ns indicates *p* > 0.05.

**Figure 3 antioxidants-11-02093-f003:**
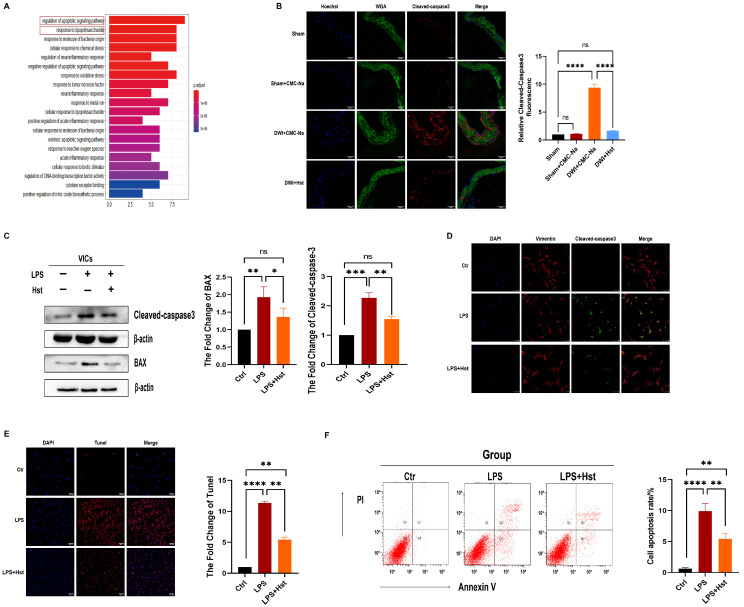
Hesperetin supplementation alleviates LPS-induced apoptosis in VICs and DWI-induced apoptosis in vivo. (**A**) GEO analysis determined that hesperetin exerts its protective effects by inhibiting cell apoptosis; (**B**) the cleaved-caspase3 protein levels (red) in mouse aortic valves were determined by immunofluorescence; scale bar = 50 μm; *n* = 5; (**C**) immunoblotting images images of cleave-caspase3 and BAX in human aortic VICs subjected to different treatments: control, LPS, and LPS + Hst and their corresponding quantification; *n* = 5; (**D**) immunofluorescence staining showed that pretreatment with hesperetin depressed the LPS-induced protein levels of cleaved-caspase3 (green) in VICs; nuclear staining was performed with DAPI (blue), and Vimentin (red) was used to show the morphology of the VICs; scale bar = 100 μm; *n* = 5; (**E**) TUNEL (red) staining of VICs under LPS or LPS + Hst stimulation; scale bar = 100 μm; *n* = 5; (**F**) the cell apoptosis of VICs was tested via flow cytometry. *n* = 5; * indicates *p* < 0.05, ** indicates *p* < 0.01, *** indicates *p* < 0.001, **** indicates *p* < 0.0001, and ns indicates *p* > 0.05.

**Figure 4 antioxidants-11-02093-f004:**
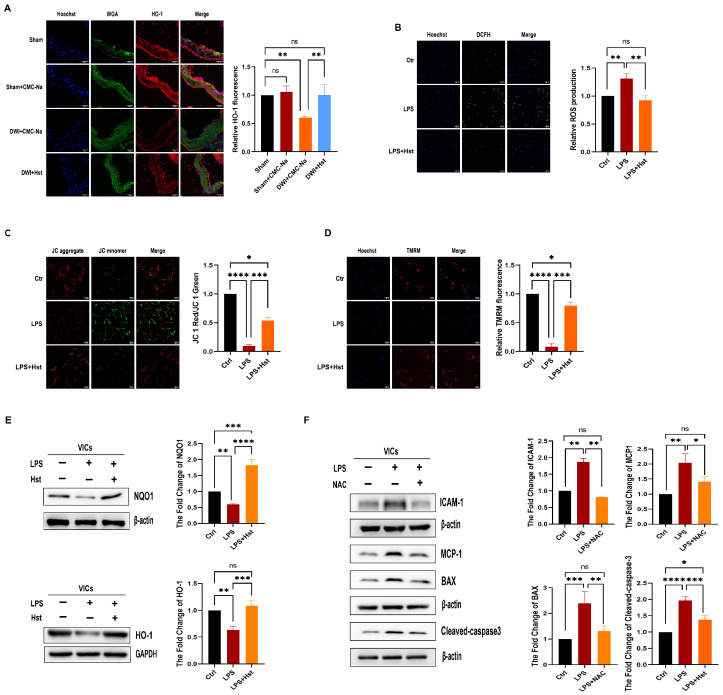
Hesperetin supplementation alleviates LPS-induced OS in VICs or DWI-induced apoptosis in vivo, and LPS-induced inflammation and apoptosis in VICs are dependent on ROS. (**A**) the HO-1 protein levels (red) in mouse aortic valves were detected by immunofluorescence staining, nuclear staining was performed with Hoechst (blue), and WGA (green) was used to show the morphology of the valve leaflet; scale bar = 50 μm; *n* = 5; (**B**) the reactive oxygen species (ROS) levels (green) in mouse aortic valves were detected with immunofluorescence, and nuclear staining was performed with Hoechst (blue); scale bar = 50 μm; *n* = 5; (**C**,**D**) fluorescence images of VICs stained with JC-1 and TMRM; scale bar = 100 μm; *n*= 5; (**E**) the levels of NQO1 and HO-1 were determined by immunoblotting; *n* = 5; (**F**) representative images show that NAC treatment reduced the ICAM-1, MCP-1, BAX, and cleaved-caspase3 levels in VICs, *n* = 5; * indicates *p* < 0.05, ** indicates *p* < 0.01, *** indicates *p* < 0.001, **** indicates *p* < 0.0001, and ns indicates *p* > 0.05.

**Figure 5 antioxidants-11-02093-f005:**
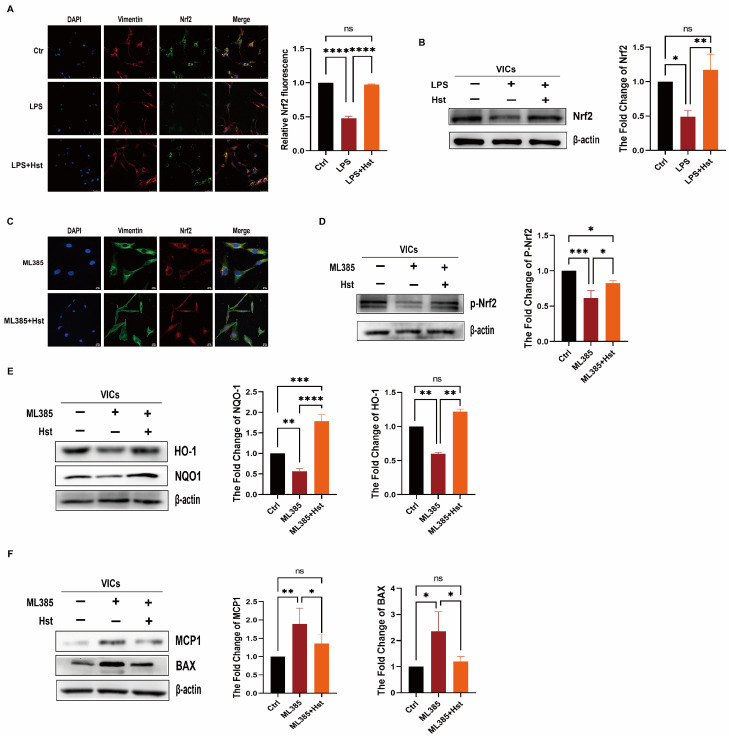
Hesperetin supplementation upregulates the Nrf2–ARE signaling. (**A**,**B**) human aortic VICs were divided into three treatment groups: control, LPS, and LPS + Hst. The Nrf2 protein levels (green) in human aortic VICs were determined via immunofluorescence, nuclear staining was performed with DAPI (blue), and Vimentin (red) was used to show the morphology of the VICs; scale bar = 75 μm; *n* = 5; and immunoblotting, *n* = 5; (**C**) representative images show the extranuclear localization of Nrf2 (red) in VICs stimulated with ML385 and the intranuclear localization of Nrf2 after human VICs were treated with hesperetin; scale bar = 20 μm; (**D**–**F**) representative immunoblotting images and quantification of the levels of p-Nrf2, NQO1, HO-1, MCP1, and BAX in VICs treated or not treated with ML385 and Hst (*n* = 5); * indicates *p* < 0.05, ** indicates *p* < 0.01, *** *p*< 0.001, **** indicates *p* < 0.0001, and ns indicates *p* > 0.05.

**Figure 6 antioxidants-11-02093-f006:**
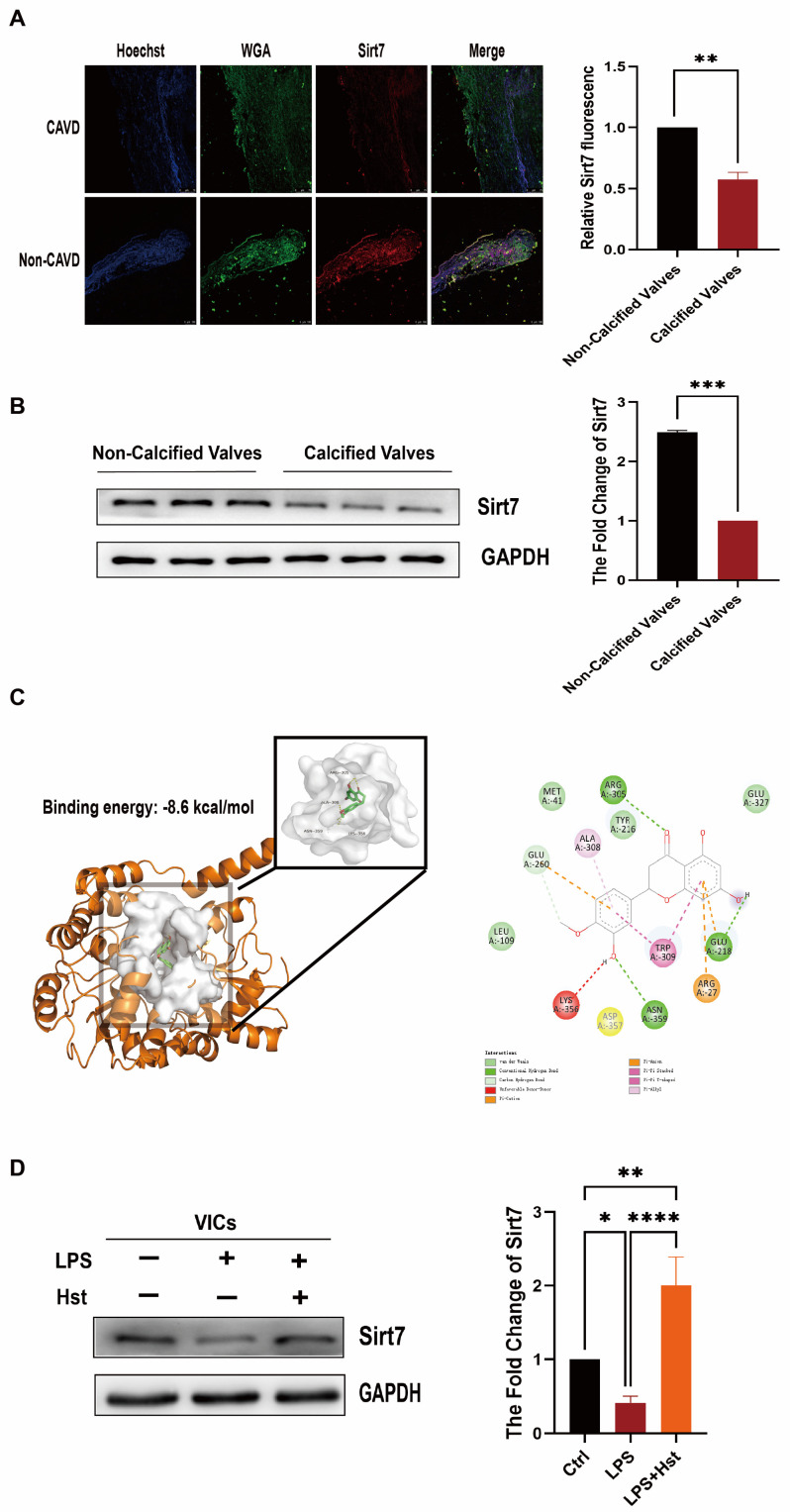
Sirt7 is involved in the protective effects of hesperetin in VICs. (**A**) the protein expression of Sirt7 (red) in CAVD patients and non-CAVD patients was detected via immunofluorescence, nuclear staining was performed with Hoechst (blue), and WGA (green) was used to show the morphology of the valve leaflet; scale bar = 100 μm; *n* = 3; (**B**) representative immunoblots and corresponding quantification of Sirt7 in human aortic VICs, *n* = 3; (**C**) docking pose of hesperetin (carbon atoms in green) into the human Sirt7 binding pocket. The key residues are displayed as sticks and colored gray. Hydrogen bonds are displayed with red dashed lines; (**D**) VICs were cultured under three different conditions (control, LPS, and LPS + Hst), and the expression of Sirt7 was then detected by Western blotting. * indicates *p* < 0.05, ** indicates *p* < 0.01, *** *p* < 0.001, **** indicates *p* < 0.0001 and ns indicates *p* > 0.05.

**Figure 7 antioxidants-11-02093-f007:**
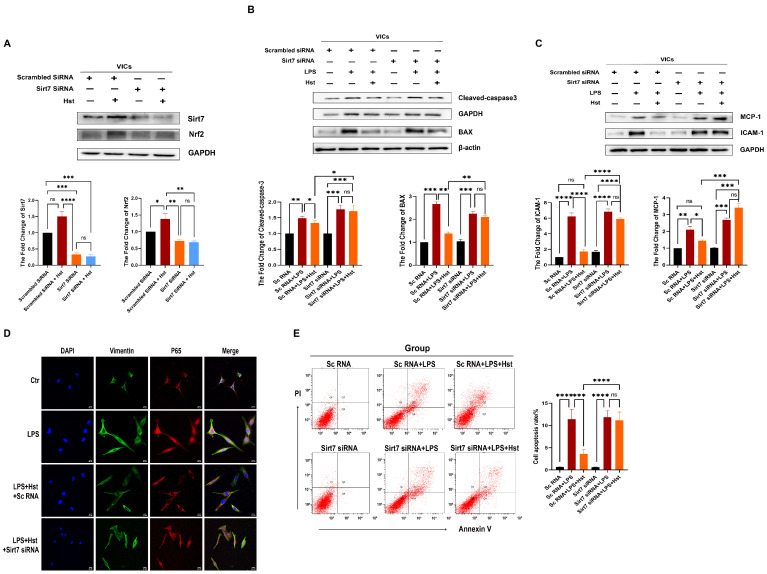
Sirt7 inhibits inflammatory responses and apoptosis by activating Nrf2 in VICs. (**A**) protein levels of Sirt7 and Nrf2 in VICs transfected with siRNA targeting Sirt7; *n* = 5; (**B**,**C**) the protein levels of cleaved-caspase3, BAX, MCP-1, and ICAM-1 in LPS and hesperetin-supplemented human VICs in which Sirt7 was knocked down were measured by Western blotting. A scrambled siRNA was transfected in VICs as a control; *n* = 3; (**D**) immunofluorescence staining confirmed that hesperetin inhibited the LPS-induced cell nuclear translocation of P65 (red) and that the knockdown of Sirt7 abolished this effect, nuclear staining was performed with DAPI (blue), and Vimentin (green) was used to show the morphology of the VICs; scale bar = 20 μm; (**E**) the apoptosis of VICs was tested via flow cytometry; *n* = 3; * indicates *p* < 0.05, ** indicates *p* < 0.01, *** indicates *p* < 0.001, **** indicates *p* < 0.0001, and ns indicates *p* > 0.05.

**Figure 8 antioxidants-11-02093-f008:**
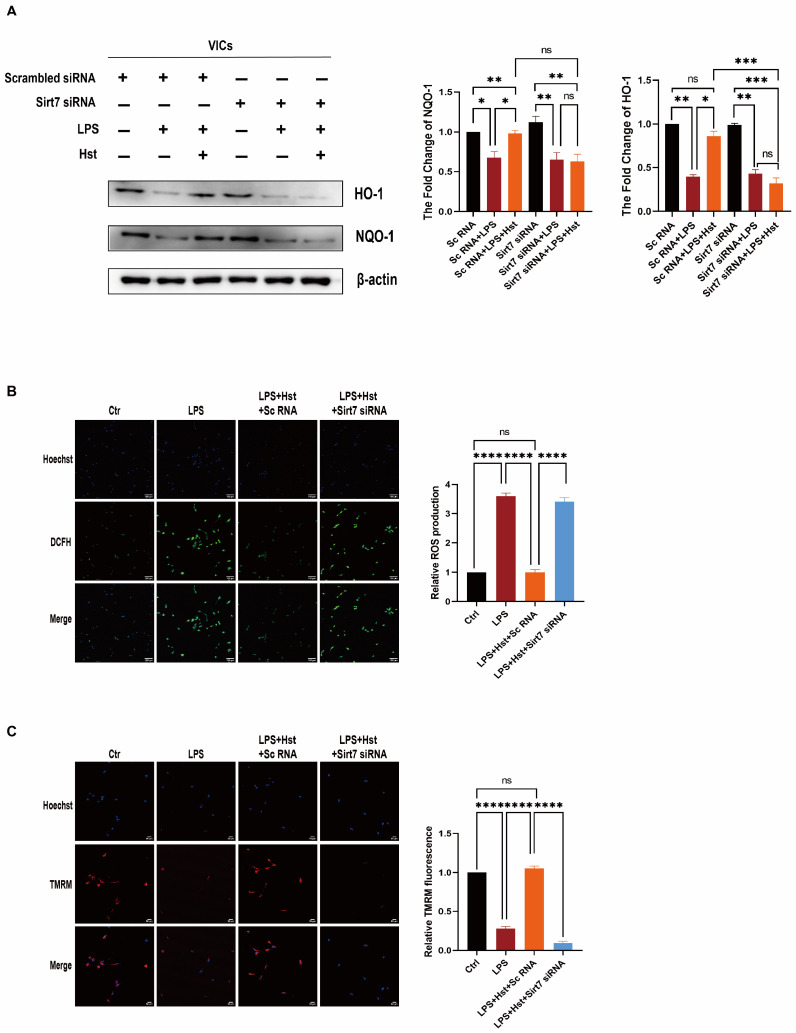
Sirt7 inhibits oxidative response by activating Nrf2 in VICs. (**A**) the protein levels of HO-1 and NQO-1 in LPS and hesperetin supplemented human VICs in which Sirt7 was knocked down were measured by Western blotting. A scrambled siRNA was transfected in VICs as a control; *n* = 3; (**B**) DCFH-DA (green) staning showed ROS production, *n* = 4; scale bar = 50 μm; (**C**) TMRM (red) staining showed MCMP function, *n* = 4; scale bar = 50 μm; * indicates *p* < 0.05, ** indicates *p* < 0.01, *** indicates *p* < 0.001, **** indicates < 0.0001, and ns indicates *p* > 0.05.

## Data Availability

The data that support the findings of this study are available from the corresponding author, Qingchun Zeng, upon request.

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
