# Peer review of "Hesperetin, a Promising Dietary Supplement for Preventing the Development of Calcific Aortic Valve Disease"

_antioxidants, 2022, doi:10.3390/antiox11112093_

Round 1

Reviewer 1 Report

Hengli Zhao et al investigated the effect of hesperetin on aortic valve stenosis induced by direct wire injury in vivo, and LPS-induced inflammation in valve interstitial cells (VICs) in vitro. They showed that hesperetin supplementation protected against wire injury-induced valve stenosis. They found that hesperetin suppressed LPS-induced activation of NF-κB and inflammatory cytokine secretion, reduced ROS production and apoptosis, and abrogated LPS-mediated injury to the mitochondrial membrane potential and decline of antioxidant levels in VICs. They showed that Nrf2 pathway and Sirt7 are involved in the protective effects of hesperretin.  

The paper is nicely written and the results are clearly presented. Although there is substantial gap between the in vitro and the in vivo models used in this work (wire injury vs. LPS).

Major issues to address:

Were the primary VICs characterized after isolation? How?

How much LPS was used in the experiments?

The bar graphs are too small, very hard to see the labelings.

It is quite surprising that LPS induces calcification of VICs within 3 days (Figure 2G). Alizarin red staining protocol is missing. The authors should provide further evidence that LPS induces osteogenic differentiation and calcification of VICs. Suitable markers are Runx2, alkaline phosphatase, osteocalcin, Ca measurement, etc.

Author Response

Dear Reviewer, 

Thanks for your comments on our manuscript “Hesperetin, a promising dietary supplement for preventing the development of calcific aortic valve disease" (Antioxidants-1909578). We carefully read the questions and comments proposed by the reviewers and made the following response.

Sincerely,

Qingchun Zeng, M.D., Ph.D.

Reviewer 2 Report

This study, “Hesperetin, a promising dietary supplement for preventing the development of calcific aortic valve disease,” has interesting and relevant concepts. Aortic valve stenosis is currently known to have a significant social and economic burden. With no pharmacological interventions available, focusing on preventive therapy is an important aspect. Here the author discusses the protective role of Hesperetin with relevant results and discussion. The authors should revise the writing, and adding some calcification-related markers to the study is strongly recommended.

·      The authors present many results and discuss them. Providing too much information can overshadow the main meaning to be conveyed, leading to mixed and confusing interpretations for readers. 

·      The quality of some figures is not good enough to see the results. 

·      The presentation of figures is not uniform; some figures are very small to see clearly. The authors should consider presenting the figures uniformly with better quality.

·      The figure legends have some errors (for example, figure 1, line 185), and the presentation is not uniform, so the legends should be revised carefully. In figure 3, what is the meaning of apoptoscheme3?

·      The authors sacrificed the mice on 8 weeks. Did the author periodically check aortic valve area and velocity before deciding or it was based on any other experiment results? In the case of DWI induced aortic stenosis, the aortic valve area and velocity decline up-to 16 weeks.

·      The authors used a 50 mg/kg dose of hesperetin, did the reference that the author cited use the same model of mice as in this experiment? The dose of the compound is very crucial, so the authors should mention why this specific dose was taken if the model of mice is different.

·      In section 3.1, line 176 author writes that H&E staining is performed to determine the degree of calcification, but results are discussed only about aortic valve area and velocity. In images, no any dark blue colors are visible, indicating calcification. This should be considered by the authors.

·      The author presented alizarin red s staining in VICs for calcium deposition, since this paper deals with calcific aortic stenosis, the authors should consider performing experiments to see the markers of calcification RUNX2, OPN, BMP2, etc. 

·      In section 3.4, too much information is provided in one section, it will be better to present LPS induced inflammation and apoptosis in VICs are ROS dependent as a new section.

·      Longer exposure periods of Hesperetin and LPS should be attempted if feasible. Exposure to LPS also induces osteogenic markers, which also should be presented. 

·      GOBP analysis and KEGG pathway analysis were performed, are there any results mentioning the involvement selected pathway (Nrf2-ARE signaling pathway)?

·      The authors presented the involvement of Sirt7 in the mechanism of CAVD. How did the authors select Sirt7? Were there any gene expressions found in gene analysis?

Author Response

(The authors gave the same response as above.)

Reviewer 3 Report

In the current study the authors utilized a mouse CAVD model induced by direct wire injury (DWI) and investigated beneficial roles of hesperetin. They found that both pre- or posttreatment of hesperetin could attenuate valve stenosis.  In cultured human VICs, hesperetin protected against lipopolysaccharide (LPS)-induced cell apoptosis, inflammation and mitochondrial membrane potential, as well as suppressed NF-κB activation and ROS generation. The authors further proposed that the possible mechanism underlying these benefits of hesperetin is through the activation of Sirt7-Nrf2-ARE pathways. This is an interesting study which provides the new evidence of hesperetin as potential supplementation for cardiovascular health. There are however some concerns:

1.       Since there are currently no therapeutic drugs on CAVD especially established CAVD, the finding of protective effect of hesperetin on valve stenosis even postoperative treatment is striking. This intriguing effect should be further validated by in vitro experiments (ie treatment of VICs with hesperetin after a period of LPS stimulation).

2.       Considering the in vivo CAVD model is induced by DWI, to better extrapolate in vitro experimental results, the authors should justify why the VICs was stimulated with LPS rather than conventional osteogenic medium?

3.       Methods: it is not very clear how hesperetin was administrated in mice? There is also lack of description how ROS was measured.

4.       Fig 1: Are there any changes of calcium deposit within mouse aortic valve leaflets? If so, please provide representative images of histology with quantifications, or any classic calcific markers such as Runx2 or OPN; Fig 1B: the peak velocity in DWI+Hst group looks unchanged or even higher compared to DWI+CMC-Na; Fig 1G: there seems also no clear difference of peak velocity between Sham+CMC-Na and DWI+CMC-Na.

5.       Since Hst itself could activate Nrf2 and Sirt7 per se (Fig 7A), some in vitro studies should include Hst treatment alone (without LPS) as another control, such as Fig 2E, Fig 4 and Fig 5.

6.       Fig 3C: please provide Western blots for cleaved-caspase3. It would be ideal to include Bcl-2 as well, and calculate the Bax/Bcl-2 ratio.

7.       Fig 5: will inhibition of Nrf2 activation with ML385 protect against LPS-induced calcium deposition of VICs? Similar question applies to Fig 7: will knockdown Sirt7 with siRNA be able to attenuate LPS-induced calcium deposition of VICs?

8.       The MS should have been carefully proofread, for example, there are some errors in Fig 3 legend; Ref 6 and 37 are repeated etc.

Author Response

(The authors gave the same response as above.)

Reviewer 4 Report

I read with interest the article “Hesperetin, a promising dietary supplement for preventing the development of calcific aortic valve disease”.

The authors evaluated the role of Hesperitin, an extract of citrus fruits, on calcific aortic valve disease by virtue of its antioxidant and anti-inflammatory role. For this purpose, they used a mouse model by direct wire injury and an in vitro culture of valve interstitial cells.

Interestingly, by this study, authors highlighted a direct involvement of Sirt7 with Hesperitin therapeutic properties.

Article is generally well written (correct in line 393: “Moreover, in vivo experiments, the expression…” and line 396: “..in both the initiation and propagation phases….”) and data are well showed and described.

Histograms: In general, all histograms are too small and some of them look squashed. Increase the size of all histograms.

In Figure 7 B and C the name of the columns is missing.

Fluorescence images: There are no scale bar, add them in all images.

The images are small, increase the size.

Add the scale bar also in figure 1C.

Captions:

Figure2: title in bold is missing

Figure 3: Check title, add description A and B

Figure 4: title in bold

Figure 5: The histogram relative to Nrf2 should be placed in box B. The histogram p-Nrf2 I think is relative to box C and the description of the columns should be Ctrl, ML385, Hst. Correct the caption.

Figure 7: Correct the caption

Figure 8: Check Title.

Materials and methods:

Add a paragraph to describe the method used for Alizarin Red staining.

Describe how the immunofluorescence and immunoblotting images were analysed.

Immunoblots:

Could you motivate why you decided to use different housekeeping (GAPDH and βactin) for the same proteins?

Paragraph 3.6:

The paragraph talks about patients and controls. It is not clear whether Sirt7 expression was evaluated on tissue isolated from DWI and control mice. What do you mean by patients and controls?

Line 215: the concentration of IL-6 is missing

Line 369: ".....LPS-stimulated conditions (Figure 8A)."

The term “pretreated”, which describes the treatment with Hst as a preventive, was used in several places in the article. For example: line 216 "In contrast, VICs pretreted with ..."; line 271 "... after VICs were pretreated with ....". However, the administration of Hst to VICs appears to be posterior to LPS. Could you add in the material and methods section a paragraph to describe the protocol of VICs treatments (LPS, siRNA, Hst)?

If it was mainly evaluated the restorative role of Hst on CAVD, the experiments conducted on mice (Paragraph 3.1) also suggest a preventive role of this molecule. Please add into discussion.

I think it is important to add the reason why Hesperitin was chosen rather than Hesperidin, and hints on its ability to cross the intestinal barrier (since the oral route was chosen for administration).

Author Response

(The authors gave the same response as above.)

Reviewer 5 Report

The manuscript by Zhao and colleagues tries to demonstrate the role of Hesperetin on aortic valve disease through Sirt 7 and Nrf2 signaling. Although the paper is interesting and potentially can contribute to a novel strategy to prevent aortic stenosis progression, the authors should clarify some weaknesses in the methodological design and conclusions of the experiments. In addition, some points of the manuscript are not well developed or explained.

The story, in general, is very confusing and not at all easy to read. Authors should make some conclusions for each block of experiments that allows for interpreting the next set of experiments. The rationale for exploring the role of Sirt 7 is unclear; somehow, it seems forced.

Specific questions:

1)     Are VICs isolated From healthy donors? Are they pooled from different donors, or VICs From individual donors are used each time? If so, how do they assess variability between donors? Also, why not prove the efficacy of Hesperetin treatment in VICs isolated from CAVD patients?

2)     The experimental model used is very aggressive and logically exerts a huge inflammatory effect. However, the hematoxylin-eosin staining only shows stenosis. The authors should add fibrosis and calcification data. The title of the first results heading, line 165, is inexact “Hesperetin supplementation protected the heart against wire injury-induced valve stenosis.” The authors do not show any protective effect on the heart. They show valve thickening and aortic peak velocity; they either must correct that title or show the impact on the heart. In the experiments shown in fig 1, authors should show aortic valve calcification since Sirt7 effects are demonstrated in CAVD, to make both studies comparable. Have the authors used other aortic valve disease model? Doing so would reinforce the results presented. 

3)     In general, appropriate controls are missing. In Figures 1 through 6, the authors compare the effect of LPS vs. LPS+ Hesperetin; since LPS+H seems to reduce the levels of some of the proteins studied below CT levels, the effect of Hesperetin alone must be shown.

4)     Figure 7 shows the effect of Si RNA scramble with LPS and LPS+Hesperetin but not Hesperetin by itself, either with SiRNA scramble or siRNA Sirt 7. Those controls are essential.

5)     The quality of the images, especially from fig 1 to 6 is very poor, and graphs are tiny and difficult to read. Some IF images have been quantitated. Please, include quantitation for all figures.

6)     Fig 7B, an explanation of color bars is missing.

7)     In some figures, WGA or vimentin staining is shown, but there is no reference in the text or figure legend. In addition, nuclear staining such as DAPI o Hoetch will be necessary to interpret some IF figures.

8)     Authors should referential the studies used to perform the KEGGS pathway analysis.

9)     Also, the conclusions of the study are overinterpreted.Nrf2 seems to be involved, but the results did not reflect the findings, and the link to Sirt 7 is relatively weak. The role of Nrf2 presented in figure 5-8 should be analyzed by Nrf2 RNA silencing and or by exploring keap 1 levels for example.

10)  A graphical abstract could help to understand and unify all the players involved in Hesperetin effect.

Minor

Authors abbreviated mitochondrial membrane potential as MMP. However, MMP universally stands for metalloproteinase and it can be confusing. It would be better if they shortened it in another way.

Author Response

Dear Reviewer, 

Thanks for your comments on our manuscript “Hesperetin, a promising dietary supplement for preventing the development of calcific aortic valve disease" (Antioxidants-1909578).

Thanks for your professional advice sincerely. We are very sorry for the confusion caused by our typography issue. For the issue, you mentioned some weaknesses in the methodological design, and we would like to explain our project design. Firstly, the main meaning of our work is to investigate the effect of hesperetin on CAVD. To achieve this goal, we first examined the effect of hesperetin in DWI mice models (Figure 1). Having demonstrated its protective roles in vivo, we then searched the databases to reveal its specific potential molecular targets and pathways. It is found that hesperetin's effects are related to inflammation, oxidative stress and apoptosis, and these pathological mechanisms are closely related to CAVD. And then, we discussed its protective effect from the above three aspects respectively (Figure 2-4). Next, we sought to discover the target of hesperetin from the bottom to up (Figure 5-8) and finally confirmed that hesperetin plays multiple protective roles in the aortic valve through the Sirt7-Nrf2-ARE axis. We have added conclusions for each block of experiments as your suggestion. And the reason why we explored the role of Sirt7 is that we firstly demonstrated hesperetin could activate the Nrf2-ARE signaling pathway, and then we have gone through the literature and discovered that the sirtuin family is an upstream regulator of Nrf2[1-3]. Therefore, we examined all sirtuin factors (Supplemental Figure ) and found that only Sirt7, reduced by LPS stimulation before, was significantly increased in abundance under hesperetin supplementation. Next, we performed the docking analysis to demonstrate the hesperetin directly binding to Sirt7. Admittedly, we have to admit that there are still some weaknesses in our research, and we will revise the article as much as possible according to your requirements, so as to present more reasonable and complete results to the readers.

We carefully read the questions and comments proposed by the reviewers and made the following response.

Sincerely,

Qingchun Zeng, M.D., Ph.D.

Round 2

Reviewer 1 Report

The manuscript improved a lot during the revision, although the quality of figures is still low. Bar graphs are still too small, graph and font sizes vary within the figures. It must be improved. 

Author Response

(The authors gave the same response as above.)

Reviewer 2 Report

The authors presented appropriate answers to the points pointed out, and the first draft of the manuscript was well revised accordingly.

However, it is recommended that English correction be performed before the final publication.

Author Response

(The authors gave the same response as above.)

Reviewer 3 Report

The MS has been significantly improved. Here are however a few more suggestions:

1.       Since the authors only showed the preventive effect of hesperetin on LPS-induced VIC calcification in vitro, as well as in line with the title of this study (“preventing”), the in vivo part about the effect of postoperative treatment of hesperatin on established stenosis could be excluded.

2.       Fig 3C: usually, it would be better to show the ratio of Westerns for cleaved-caspase 3/total caspase 3; Fig 3D: in compliance with the in vivo study (Fig 3B), it would be good to show the staining of cleaved caspase 3 in VICs rather than total caspase 3.

3.       Fig 4F: again, please provide Westerns for cleaved caspase-3 if possible.

Author Response

(The authors gave the same response as above.)

Reviewer 4 Report

The manuscript is much clearer after the corrections made.

Please check line 59

line 317 correct "cleaved"

The figures have not been enlarged and the writings continue to be difficult to read.

Author Response

(The authors gave the same response as above.)

Reviewer 5 Report

The authors had made an effort to improve the quality of the images and the manuscript. The results are more clearly presented. Most of my questions have been answered.

Author Response

(The authors gave the same response as above.)

Round 3

Reviewer 1 Report

No further comments.

Reviewer 4 Report

The manuscript was implemented.